# Hearing Problems in Indonesia: Attention to Hypertensive Adults

**DOI:** 10.3390/ijerph19159222

**Published:** 2022-07-28

**Authors:** Melysa Fitriana, Chyi-Huey Bai

**Affiliations:** 1International Master/Ph.D. Program in Medicine, College of Medicine, Taipei Medical University, Taipei 11031, Taiwan; melysa.fitriana@mail.ugm.ac.id; 2Otorhinolaryngology Head and Neck Surgery Department, Faculty of Medicine, Public Health and Nursing, Universitas Gadjah Mada, Yogyakarta 55281, Indonesia; 3School of Public Health, College of Public Health, Taipei Medical University, Taipei 11031, Taiwan; 4Department of Public Health, College of Medicine, Taipei Medical University, Taipei 11031, Taiwan

**Keywords:** hearing problem, hypertension, adult, IFLS

## Abstract

Known as a silent disability, hearing loss is one of the major health burdens worldwide. Evidence implies that those suffering from hypertension can experience hearing disturbances. Self-reporting of hearing problems and self-reporting of hypertension may be useful in providing an alarm for detecting hearing problems. However, in the Indonesian population, this matter has not been properly reported. The aim of this study was to explore the prevalence of hearing problems and their relationships with other demographic factors. In total, 28,297 respondents of productive age from the Indonesian Family Life Survey 5th wave were assessed. A questionnaire and physical examination data were included in this survey. Self-reported hearing problems and their predictors were analyzed using univariate and multivariate logistic regressions. Hypertension awareness was a significant predictor of having a hearing problem (odds ratio (OR) [95% confidence interval (CI)], *p* value: 2.715 [1.948~3.785], <0.001). Having a general check-up was also crucial for detecting hearing problems (2.192 [1.54~3.121], <0.001). There was a significant link between hearing problems and early adults who have isolated systolic hypertension. Hypertension awareness and having a general check-up had predictive value for detecting hearing problems in adults in the age range of 26~35 years. Therefore, public health strategies for hearing loss prevention might target this group by detecting and treating hypertension.

## 1. Introduction

The sense of hearing is important for humans to communicate with other people and also to interact within the community. With advancements in hearing technologies in the past few decades, identifying and diagnosing hearing impairment can be completed in many settings. Around 360 million people (over 5%) all over the world live with this disability (disabling hearing loss), which makes hearing loss a factor that is responsible for years lived with a disability [1,2]. Among the six World Health Organization (WHO) regions, the highest prevalence of hearing loss is in the western Pacific region (with 136.5 million people), followed by Southeast Asia (with 109.5 million people) [3].

Loss of productivity and social isolation might result from communication difficulties caused by hearing impairment [4,5]. A systematic review of the global burden of disease from 1990 to 2019 revealed that 1.57 billion people worldwide suffered from hearing impairment in 2019. An estimate was made that by 2050, there will be 2.45 billion people with a hearing decline (one of every four people), an increase of 56.1% from 2019 [6]. Although there are approximately 40 million American adults who suffer from various degrees of hearing impairment, there are gaps between self-reported hearing loss and people who receive a hearing test and treatment for hearing impairment [7]. Patients who have self-recognized hearing disturbances should undergo a hearing evaluation by a medical professional such as an ear, nose, and throat (ENT) doctor or audiologist [8].

Based on the National Health and Nutrition Examination Survey in 2015–2018, the prevalence of high blood pressure increases with age, at 28.2% among 20~44 year olds, at 60.1% in 45~64 year olds, and 77% among those older than 65 years [9]. On the other hand, low- and middle-income countries (LMICs; with 1.04 billion people accounting for 31.5% of the world’s population) had a higher prevalence of hypertension compared to high-income countries (HICs; with 349 million people or 28.5% of the world’s population) [10,11]. Unfortunately, in spite of the increasing hypertension prevalence, hypertension awareness and treatment in LMICs are still low, including in Indonesia, which had treatment rates of less than 25% of women, as well as men which less than 20% [10,12]. For several decades (1990~2019) the hypertension burden has increased in young adults [13]. In Asian populations, the prevalence of isolated systolic hypertension (ISH) has increased due to special epidemiological characteristics and risk factors and greater susceptibility to ISH [14]. Likewise, a previous study also implied that systolic blood pressure (SBP) levels are considered a crucial factor in preventing and treating hypertension [15].

Epidemiological studies of hearing loss and hypertension as vascular risk factors supported the fact that there is a significant link between the two [16,17]. However, the molecular mechanism behind this association is still under investigation, and it could be due to vascular injury [18,19]. A recent cohort study investigating hearing loss and blood pressure was conducted in Japanese adults, and those findings suggested that a higher SBP condition was a significant risk factor for hearing loss at 1 kHz [3]. A long time ago, the crucial role of SBP was already mentioned in the Framingham Heart study which determined that SBP had a larger capacity in the future risk of cardiovascular disease than diastolic blood pressure (DBP) [20,21].

In this study, our main aim was to investigate the prevalence of hearing problems of various ages and blood pressure conditions. Later, we assessed the relationship between hearing problems and blood pressure based on subjects’ age groups in an Indonesian population.

## 2. Materials and Methods

### 2.1. Study Population and Participants

The Indonesian Family Life Survey 5th wave (IFLS5) 2014 was used in this study. The IFLS5 is a cross-sectional national survey that focuses on health and socioeconomic fields and was conducted from late 2014 to early 2015. Data were collected by assessing individual respondents, their households, and community levels. The IFLS survey is a longitudinal survey designed to provide public use data to investigate behaviors and outputs, and to the present, there have been five waves of surveys that are available on their website. In the first wave in 1993, 20 households from urban regions and 30 households from rural regions in 13 of 27 provinces in Indonesia were selected by random sampling. IFLS5 was chosen as the data source in this study because it has the most complete data compared to previous waves. A well-designed structured questionnaire was administered through in-person interviews by well-trained interviewers. A physical examination including blood pressure, body height, and body weight was also conducted [22]. Our study population included all of the participants of the IFLS5, and the sample consisted of those who met the inclusion and exclusion criteria. The inclusion criterion was an age range of 15~64 years, and exclusion criteria were missing data of gender, blood pressure, height, or weight. Ultimately, 28,297 participants were enrolled as seen in Figure 1.

### 2.2. Age Categories

Respondents’ ages ranged from 15 to 64 years, which is considered productive age in the Indonesian population. Furthermore, age was divided into four groups based on Indonesia Ministry of Health criteria: (1) 15~25 years as adolescence, (2) 26~35 years old as early adulthood, (3) 36~45 years old as late adulthood, and (4) 46~64 years old as elderly.

### 2.3. Blood Pressure

Blood pressure was measured three times by trained interviewers using an Omron meter (HEM-7203) on alternate arms in a seated position. A normal size cuff was generally used, while larger cuffs were also provided if needed. Averages of the systole and diastole data were determined. Later, blood pressure data were classified using the modified-American Heart Association (AHA) 2017 and Indonesian Society of Hypertension (INASH) 2019 criteria. AHA 2017 [23] criteria for blood pressure were modified into three groups: (1) normal blood pressure (systole < 120 mmHg and diastole < 80 mmHg), (2) elevated blood pressure (systole 120~129 mmHg and diastole < 80 mmHg) and (3) hypertension (systole ≥ 130 mmHg and diastole ≥ 80 mmHg).

To evaluate specific cases of ISH, additional criteria of the INASH 2019 with modifications were also added. The modified INASH version only has two groups which are non-ISH (a combination of optimal, normal, high normal, and grades 1, 2, and 3 hypertensive groups) and ISH. Originally, the INASH criteria had several groups of blood pressure, which included (1) optimal (systole < 120 mmHg and diastole < 80 mmHg), (2) normal (systole 120~129 mmHg and diastole 80~84 mmHg), (3) high-normal (systole 130~139 mmHg and diastole 85~89 mmHg), (4) grade 1 hypertension (systole 140~159 mmHg and diastole 90~99 mmHg), (5) grade 2 hypertension (systole 160~179 mmHg and diastole 100~109 mmHg), (6) grade 3 hypertension (systole ≥ 180 mmHg and diastole ≥ 110 mmHg), and (7) ISH (systole ≥ 140 mmHg and diastole < 90 mmHg) [24].

### 2.4. Hearing Problems-Self Reported

Hearing problems were assessed by answering a question about hearing problems (have you been diagnosed with a hearing problem by a doctor, paramedic, nurse, or midwife?).

### 2.5. Covariates

Covariates consisted of hypertension-self reported, hypertension medication, body mass index (BMI), educational level, occupation, general check-ups, outpatient care, insurance coverage, and hearing aid usage. Information on hypertension self-reported was collected by answering a question about a diagnosis of hypertension (yes or no). Likewise, hypertension medication was reported by answering a question about a history of undergoing hypertension treatment (yes or no).

The BMI was calculated using height divided by weight squared (kg/m^2^) from physical measurements. A Seca model 213 plastic height board was used to measure height to the closest millimeter. A Camry model EB1003 scale was used to calculate body weight to the nearest 0.1 kg. Finally, the BMI was categorized into four groups based on Indonesia Ministry of Health criteria, which are (1) underweight (BMI < 18.5 kg/m^2^), (2) normal (BMI 18.5~25 kg/m^2^), (3) overweight–mild (BMI 25.1~27 kg/m^2^), and (4) overweight–severe (BMI > 27 kg/m^2^).

Educational data were grouped into three categories, including (1) no schooling, (2) senior high school or lower (graduated), and (3) above senior high school (graduated). The occupational status was noted with questions about whether the respondent had worked within the past year and had the same job for more than 5 years. General check-ups, outpatient care, insurance, and hearing aid usage were recorded by answering questions of having had a general check-up in the last 5 years, having had outpatient care in the last 4 weeks, having a health card, and ever having to wear a hearing aid (all yes or no), respectively.

### 2.6. Statistical Analysis

Descriptive analyses were used to show characteristics of study participants. The frequency, percentage, chi-squared test, or Fisher exact test for categorical data and the mean, standard deviation (SD), and Student’s *t*-test for continuous data were used. The odds ratio (OR), 95% confidence interval (CI), and *p* value were calculated using univariate and multivariate logistic regressions. A subgroup analysis was carried out by gender. All statistical analyses were performed with SPSS software vers. 26 (IBM, Armonk, NY, USA). A *p* value of <0.05 was accepted as statistically significant.

## 3. Results

### 3.1. Characteristics of the Population

Of the 28,297 total individuals (age (mean ± SD): 35.29 ± 12.78 years for men and 35.01 ± 12.79 years for women) from the Indonesian population, 220 people had hearing problems (for a prevalence of 7.8 per 1000 persons). The prevalence of self-reported hearing problems in those aged 15~25, 26~35, 36~45, and 46~64 years was 0.48, 0.48, 0.85, and 1.4 per 1000 persons, respectively. Thus, the prevalence of hearing problems increased with age. In addition, hearing problems were more prevalent among subjects with elevated blood pressure or hypertension, those with isolated hypertension, those taking antihypertensive drugs, those with self-reported hypertension, those employed, those with a general check-up in the past 5 years, and those with outpatient treatment in the last 4 weeks. Among those who were experiencing hearing problems, 51.4% were male and 48.6% were female, but none of them were using a hearing aid.

Overall, the hypertensive group comprised a majority of the population according to AHA 2017 criteria which was 12,843 participants. The prevalence of hypertension based on AHA 2017 criteria was 45.38 per 100 persons in this Indonesian population aged 15~64 years. Hypertension was not significantly correlated with hearing problems (*p* = 0.093). According to the criterion of the INASH 2019, the prevalence of isolated systolic hypertension was 6.3 per 100 persons.

Those participants with hearing problems had a significantly (*p* = 0.048) higher proportion of isolated hypertension (9.5%) than those without such problems (6.3%). Surprisingly, among those with hearing problems, only 26.8% were aware that they had hypertension and 7.7% were treating it with hypertension medication.

There were correlations of hearing problems with being employed (*p* = 0.007), having had a recent general check-up (*p* < 0.001), and having had recent outpatient care (*p* = 0.003). Although the prevalence of hearing problems in the worker group at 77.3% was higher than in the non-worker group, the number of hearing problems was higher in the group who had had a general check-up in the past 5 years and those who had had outpatient care in the past 4 weeks.

Such positive correlations were not found for the BMI, educational level, or insurance ownership (Table 1).

### 3.2. Univariate Analysis of Hearing Problems with Demographic Data

Table 2 shows relationships between self-reported hearing problems as the dependent variable and other independent variables. There were significant associations (all *p* < 0.05) between hearing problems and age group, with the strongest protective effects in those aged 46–64 years. Otherwise, gender was not correlated with hearing problems (*p* = 0.144).

Based on all criteria for hypertension by AHA and INASH, significant risks were shown in the hypertension group (OR [95% CI], *p* value; 1.373 [0.019~1.851], 0.037) and in the isolated hypertension group (1.572 [1.0~2.471], 0.05) compared to the reference group. This shows that having hearing problems was related to a hypertensive condition. Likewise, self-reported hypertension (3.127 [2.315~4.224] <0.001) and related treatment (4.085 [2.473~6.748], <0.001) had large risk effects on hearing problems.

A general check-up (2.371 [1.685~3.338], <0.001) and outpatient care (1.592 [1.174~2.159], 0.003) among participants had significant risks for the occurrence of hearing problems. In addition, the underweight group based on the BMI had a significant protective effect against hearing problems (0.462 [0.249~0.857], 0.014) compared to the overweight group. Having a job seems to be related to hearing problems (1.545 [1.126~2.12], 0.007) compared to not having a job. Nevertheless, there was no relationship between hearing problems and educational level or insurance ownership (Table 2).

### 3.3. Association of Self-Reported Hypertension and General Check-Ups with Hearing Problems in Adults

Since the age and blood pressure categories were highly correlated with hearing problems, the combined effect of age group and blood pressure categories (hereafter referred to as the interaction of age*blood pressure) with hearing problems was next assessed. Subsequently, several models were developed to predict having hearing problems based on univariate *p* < 0.05 of covariates (Table 3). Compared to the full model, the general check-up and outpatient care variables were excluded in model 1. On the other hand, self-reported hypertension and outpatient care variables were excluded in model 2. In model 3 (the full model), compared to 36~45-year-old adults with normal blood pressure, hearing problems were negatively associated with participants who had elevated blood pressure in early adults (0.398 [0.165~0.959], 0.04) and the elderly (0.325 [0.11~0.961], 0.042), and were positively associated with participants who had elevated blood pressure in adolescents (2.165 [1.11~4.24], 0.023). In addition, significant positive relationships with hearing problems were found in subjects with both self-reported hypertension and having had a general check-up.

Further analysis showed that self-reported hypertension seemed to be an important predictor for having hearing problems as seen in model 1 (2.715 [1.948~3.785], <0.001). In addition, as seen in model 2, having a general check-up was a significant predictor for hearing problem conditions (2.192 [1.54~3.121], <0.001). These findings illustrate that hypertension awareness and having a general check-up are important factors; however, after adjusting for such factors, elevated blood pressure was still important to prevent hearing problems in early adults (26~35 years of age) (Table 3).

### 3.4. Protective Value of Self-Reported Hypertension and a General Check-Up to Hearing Problems in Early Adults

There was also a combined effect of age and blood pressure categories in INASH-isolated systolic hypertension which were shown in Table 4 after adjustment for covariates (gender, BMI, hypertension self-reported, general check-up, outpatient care, educational level, occupation, and insurance ownership). Compared to subjects aged 36~45 years and those without isolated systolic hypertension, there was a significant protective effect against hearing problems in early adults (26~35 years of age) (0.6 [0.394~0.915], 0.0018). The risk of hearing problems was also seen in the aged group (46~64 years) compared to those aged 36~45 years without isolated systolic hypertension (Table 4).

In addition, the subgroup analysis by gender had been performed. The result from the gender analysis showed insignificant results due to the small sample size in hearing problems (Appendix A).

## 4. Discussion

The study finding of a combined effect of age and blood pressure on hearing problems stresses the importance of the SBP condition in early adults (26~35 years of age) to prevent hearing problems. Hence, to eliminate the risk of hearing problems related to hypertension, public health professionals might target this population age range.

There was a slightly higher prevalence of hearing problems in men (51.4%) than in women (48.6%) in this study. Meanwhile, the global prevalence of hearing loss at moderate or higher levels among men is slightly higher than that of women, at 217 million (5.6%) and 211 million (5.5%), respectively [25]. Interestingly, a study conducted in 2017 showed that males were nearly two-fold more likely compared to women to have hearing impairment in populations ranging 20 to 69 years of age [26]. In contrast to a previous study in Japan, 46.7% of men and 53.2% of women had hearing impairment. Moreover, a higher risk of hearing problems occurred in both males and females with hypertension [27]. On the other hand, specifically in the male population, hypertension was positively related to hearing impairment at a prevalence ratio of 1.52 per 1000 persons with a 95% CI of 1.07~2.16 [17].

Study findings in a Canadian population showed that the prevalence of hearing problems (79%) was higher among adults with hypertension [27]. Additionally, the highest rate of hearing loss was in the group 60~69 years old [26]. Both of those findings support our finding that there was an increasing prevalence rate with age. In addition, a previous study revealed that age is known to be the strongest predictor of hearing impairment in adults (range 20~69 years old) [26]. Likewise, we found that each age group was positively associated with hearing problems (*p* < 0.05).

Based on a modified AHA 2017 grouping of blood pressure, we also found that the hypertensive group was associated with hearing problems (*p* = 0.037) as well as the modified–isolated systolic hypertensive group (*p* = 0.05). This is similar to a previous study that found that hypertension was positively associated with hearing loss. To be exact, results of audiometric examinations showed that the hypertensive group had a higher hearing capacity (23.4 ± 8.67 dB) in comparison with the non-hypertensive group (18.3 ± 6.02 dB), and the longer a person had had hypertension, the worse was their hearing ability [28]. Another study using audiometry also supported that a worse hearing level was found in hypertensive participants [29,30]. Emerging evidence proved that there was a gradually increasing level of a pure tone average (PTA) threshold and hearing impairment percentages at every frequency with an increase in the systolic standard deviation (*p* < 0.05) [31]. In 2019, Umesawa et al. tried to explore the mechanism behind the relationship between hearing impairment and hypertension. It was proven that bilateral hearing impairment was clearly caused by hypertension due to microvessel injury. Furthermore, they presumed that hypertension might damage both the inner ear organ and primary auditory cortex [32]. Additionally, a slightly increased risk of hearing impairment had a relationship with a history of hypertension [33].

Supporting our finding that hearing loss was associated with both age and hypertension, there is also a study that indicated that hypertension was related to age (*p* < 0.001) [34]. In addition, several lines of evidence showed that aging was positively associated with ISH, and ISH commonly occurred in young adults and the elderly [14,35,36]. We tried to make connections between those findings and other predictors. Although we observed that self-reported hypertension and having a general check-up were crucial predictors for avoiding hearing problems in adults aged 26~35 years, we did not find any previous study that supported those findings. Our findings in this study might be the first description of hearing problems being related to age and hypertension.

An increase in systolic blood pressure in early adults (26~35 years old) may be important, as it might increase the risk of hearing disturbances. So, we would encourage adults aged 26~35 years to have regular blood pressure check-ups and have early intervention if they have an elevated SBP in order to prevent hearing loss conditions. Moreover, this study emphasized the relationship between hearing problems and hypertension, and hence, this could be useful for future studies related to hypertension, especially in Indonesian populations.

Apart from the subjectivity of self-reported hearing loss and self-reported hypertension, only a few indicators related to hearing disturbances were available in this study. This survey also did not have noise exposure information. In addition, there was a discrepancy between self-reported hypertension with blood pressure measurement results, and some of the blood pressure measurements were non-responded cases. Another limitation was the respondents’ volunteer effect, which might have been a factor that could have affected the results. We explored every variable which we thought to be useful in this study; however, these limitations were considerable since the survey mainly explored social aspects.

## 5. Conclusions

There was a predictive value of self-reported hypertension (awareness) and having a general check-up against hearing problems in early adults. Thus, regularly monitoring the blood pressure can be suggested for this age group. Future studies are needed to investigate the mechanism of these findings.

## Figures and Tables

**Figure 1 ijerph-19-09222-f001:**
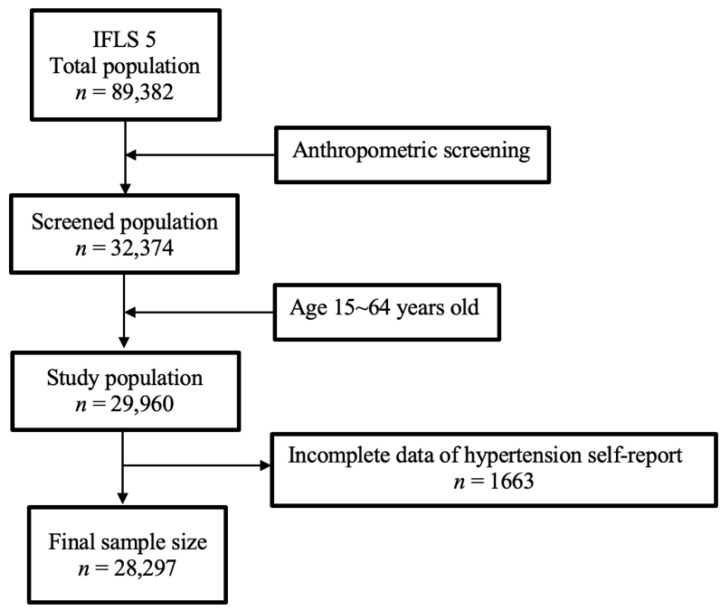
Sample screening process.

**Table 1 ijerph-19-09222-t001:** Distributions of demographic data based on hearing problems.

	*N* Total	Prevalence of Hearing Problems (%)	Hearing Problems				X^2^	*p* Value
			No		Yes			
			*n*	%	*n*	%		
Age group (years)								
15~25	7452	0.48	7416	26.4	36	16.4	50.815	0.000
26~35	8179	0.48	8139	29	40	18.2		
36~45	6193	0.85	6140	21.9	53	24.1		
46~64	6473	1.4	6382	22.7	91	41.4		
Gender								
Male	13,147	0.85	13,034	46.4	113	51.4	2.14	0.143
Female	15,150	0.7	15,043	53.6	107	48.6		
Blood pressure								
AHA 2017								
Normal	11,153	0.63	11,082	39.5	71	32.3	4.741	0.093
Elevated	4301	0.86	4264	15.2	37	16.8		
Hypertensive	12,843	0.87	12,731	45.3	112	50.9		
INAHS 2019								
Non-isolated systolic hypertensive	26,510	0.75	26,311	93.7	199	90.5	3.911	0.048
Isolated systolic hypertensive	1787	1.17	1766	6.3	21	9.5		
Hypertension medication								
Yes	581	2.92	465	2	17	7.7	35.49	0.000
No	27,716	0.73	27,513	98	203	92.3		
Self-reported hypertension								
Yes	3004	1.96	2945	10.5	59	26.8	61.34	0.000
No	25,293	0.63	25,132	89.5	161	73.2		
Body mass index group								
Underweight	3366	0.38	3353	11.9	13	5.9	7.76	0.053
Normal	15,856	0.83	15,723	56	133	60.5		
Overweight–mild	3543	0.79	3515	12.5	28	12.7		
Overweight–severe	5532	0.83	5486	19.5	46	20.9		
Education								
No schooling	7920	0.77	7859	28	61	27.7	1.88	0.39
Senior high school or lower	17,664	0.75	17,532	62.4	132	60		
Above senior high school	2713	0.99	2686	9.6	27	12.3		
Occupation								
Working	19,474	0.87	19,304	68.8	170	77.3	7.38	0.007
Not working	8823	0.57	8773	31.2	50	22.7		
General check-up								
Yes	2514	1.63	2473	8.8	41	18.6	26.05	0.000
No	25,783	0.69	25,604	91.2	179	81.4		
Outpatient care								
Yes	5015	1.12	4959	17.7	56	25.5	9.09	0.003
No	23,282	0.7	23,118	82.3	164	74.5		
Insurance ownership								
Yes	14,183	0.8	14,069	50.1	114	51.8	0.25	0.613
No	14,114	0.75	14,008	49.9	106	48.2		
Hearing aid usage								
Yes	19	0	19	0.1	0	0	0.15	0.7
No	28,278	0.78	28,058	99.9	220	100		

AHA, American Heart Association; INASH, Indonesian Society of Hypertension.

**Table 2 ijerph-19-09222-t002:** Univariate logistic regression of hearing problems and demographic data.

Predictor	Coef.	SE Coef.	Wald	*p*	Odds Ratio	95% CI Lower	95% CI Upper
Age group (years)							
15~25	−0.576	0.217	7.057	0.008	0.562	0.368	0.86
26~35	−0.563	0.21	7.185	0.007	0.569	0.377	0.859
36~45					1.0		
46~64	0.502	0.174	8.348	0.004	1.652	1.175	2.322
Gender							
Male	0.198	0.135	2.136	0.144	1.219	0.935	1.589
Female					1.0		
Blood pressure							
AHA 2017							
Normal					1.0		
Elevated	0.303	0.204	2.221	0.136	1.354	0.909	2.018
Hypertensive	0.317	0.152	4.338	0.037	1.373	1.019	1.851
INAHS 2019							
Non-isolated systolic hypertensive					1.0		
Isolated systolic hypertensive	0.452	0.231	3.845	0.05	1.572	1.0	2.471
Hypertension medication							
Yes	1.407	0.256	30.212	0.000	4.085	2.473	6.748
No					1.0		
Self-reported hypertension							
Yes	1.140	0.153	55.225	0.000	3.127	2.315	4.224
No					1.0		
Body mass index group							
Underweight	−0.771	0.315	6.001	0.014	0.462	0.249	0.857
Normal	0.009	0.172	0.003	0.959	1.009	0.72	1.413
Overweight–mild	−0.051	0.241	0.43	0.831	0.95	0.593	1.523
Overweight–severe					1.0		
Educational level							
No schooling	−0.259	0.232	1.24	0.266	0.772	0.49	1.217
Senior high school or lower	−0.289	0.212	1.854	0.173	0.749	0.494	1.135
Above senior high school					1.0		
Occupation							
Working	0.435	0.161	7.269	0.007	1.545	1.126	2.12
Not working					1.0		
General check-up							
Yes	0.864	0.174	24.511	0.000	2.371	1.685	3.338
No					1.0		
Outpatient care							
Yes	0.465	0.156	8.931	0.003	1.592	1.174	2.159
No					1.0		
Insurance ownership							
Yes	0.068	0.135	0.255	0.614	1.071	0.821	1.396
No					1.0		

AHA, American Heart Association; INASH, Indonesian Society of Hypertension; Coef., coefficient; SE, standard error; CI, confidence interval.

**Table 3 ijerph-19-09222-t003:** Multivariate logistic regression analysis of hearing problems.

Predictor		Model 1				Model 2				Full Model			
		*p*	Odds Ratio	95% CI Lower	95% CI Upper	*p*	Odds Ratio	95% CI Lower	95% CI Upper	*p*	Odds Ratio	95% CI Lower	95% CI Upper
(Constant)		0.000	0.007			0.000	0.007			0.000	0.005		
Age (years) * AHA category													
1	(15~25) * Normal BP	0.45	0.783	0.414	1.478	0.339	0.733	0.388	1.386	0.461	0.787	0.417	1.488
2	(15~25) * Elevated BP	0.335	0.648	0.268	1.567	0.288	0.619	0.256	1.499	0.325	0.642	0.265	1.552
3	(15~25) * Hypertension	0.036	0.391	0.162	0.941	0.041	0.399	0.166	0.961	0.04	0.398	0.165	0.959
4	(26~35) * Normal BP	0.1	0.584	0.308	1.108	0.084	0.568	0.3	1.078	0.103	0.587	0.309	1.114
5	(26~35) * Elevated BP	0.037	0.315	0.107	0.931	0.044	0.329	0.111	0.972	0.042	0.325	0.11	0.961
6	(26~35) * Hypertension	0.078	0.551	0.284	1.068	0.136	0.604	0.312	1.171	0.091	0.564	0.291	1.095
7	(36~45) * Normal BP		1.0				1.0				1.0		
8	(36~45) * Elevated BP	0.984	1.027	0.461	2.288	0.781	1.12	0.503	2.496	0.855	1.077	0.483	2.402
9	(36~45) * Hypertension	0.305	0.727	0.395	1.337	0.593	0.848	0.462	1.554	0.344	0.745	0.405	1.371
10	(46~64) * Normal BP	0.799	1.102	0.52	2.335	0.808	1.097	0.518	2.324	0.836	1.083	0.511	2.294
11	(46~64) * Elevated BP	0.019	2.23	1.144	4.346	0.012	2.36	1.212	4.593	0.023	2.165	1.11	4.224
12	(46~64) * Hypertension	0.562	1.174	0.683	2.016	0.094	1.568	0.926	2.654	0.562	1.174	0.683	2.018
Gender		0.108	1.273	0.949	1.709	0.365	1.146	0.854	1.537	0.128	1.259	0.936	1.692
Body mass index group													
	Underweight	0.115	0.592	0.309	1.136	0.101	0.58	0.303	1.112	0.164	0.63	0.328	1.208
	Normal	0.401	1.165	0.815	1.666	0.481	1.137	0.796	1.623	0.265	1.226	0.857	1.755
	Overweight–mild	0.936	0.981	0.61	1.577	0.848	0.954	0.594	1.535	0.974	0.992	0.617	1.596
	Overweight–severe		1.0				1.0				1.0		
Self-reported hypertension		0.000	2.715	1.948	3.785					0.000	2.48	1.77	3.474
General check-up						0.000	2.192	1.54	3.121	0.000	1.976	1.384	2.821
Outpatient care										0.066	1.343	0.981	1.839
Education													
	No schooling	0.157	0.712	0.444	1.14	0.49	0.844	0.521	1.366	0.487	0.843	0.521	1.364
	Senior high school or lower	0.266	0.788	0.517	1.199	0.586	0.888	0.58	1.361	0.604	0.893	0.583	1.369
	Above senior high school		1.0				1.0				1.0		
Occupation		0.257	1.225	0.862	1.742	0.362	1.179	0.828	1.678	0.242	1.233	0.868	1.753
Insurance ownership		0.631	1.067	0.818	1.393	0.651	1.063	0.815	1.388	0.685	1.057	0.809	1.379

AHA, American Heart Association; BP, Blood Pressure; CI, confidence interval; *, Combine effect of age and blood pressure groups.

**Table 4 ijerph-19-09222-t004:** Combination effect of age and blood pressure criteria on hearing problems.

Predictors		Coef.	SE Coef.	Wald	*p*	Odds Ratio	95% CI Lower	95% CI Upper
Age (years) * INASH-isosysht								
1	(15~25) * Non-isosysht	−0.315	0.232	1.849	0.174	0.73	0.463	1.149
2	(15~25) * Isosysht	−16.398	2764.809	0.000	0.995	0.000	0.000	-
3	(26~35) * Non-isosysht	−0.511	0.215	5.641	0.018	0.6	0.394	0.915
4	(26~35) * Isosysht	−0.667	1.014	0.433	0.511	0.513	0.07	3.745
5	(36~45) * Non-isosysht					1.0		
6	(36~45) * Isosysht	−0.36	0.724	0.246	0.62	0.698	0.169	2.887
7	(46~64) * Non-isosysht	0.337	0.19	3.156	0.076	1.401	0.966	2.032
8	(46~64) * Isosysht	0.526	0.284	3.419	0.064	1.692	0.969	2.954

Adjusted for all predictors (gender, body mass index group, hypertension self-reported, general check-up, outpatient care, educational level, occupation, and insurance ownership). Coef., coefficient; SE, standard error; CI, confidence interval; AHA, American Heart Association; BP, blood pressure; INASH, Indonesian Society of Hypertension; Non-isosysht, non-isolated systolic hypertension; Isosysht, isolated systolic hypertension; *, Combine effect of age and blood pressure groups.

## Data Availability

This research was conducted by using IFLS 5th waves provided by RAND (http://wwww.rand.org) and this database is open source. Last accessed on 1 May 2022.

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
