# Peer review of "Hearing Problems in Indonesia: Attention to Hypertensive Adults"

_ijerph, 2022, doi:10.3390/ijerph19159222_

Round 1

Reviewer 1 Report

Any early detection to prevent medical damage and that does not involve expensive tests is welcome. Thus, awareness of hypertension also in the context of hearing loss and not only in the context of heart problems is an issue that needs to focus attention and outreach in the system responsible for public health.

It should be noted that age is a factor that must be neutralized as it is associated with both high blood pressure and hearing problems. It would have been worthwhile to neutralize age in any data analysis.

Of the 239,806 cases, only 82,686 (34.5%) remained due to missing data relevant to the study. Although it is a broad scale it may be biased. 

In Table 2 it is not clear why the age group 36-45 was chosen as the reference group. Why not the 15-25 age group which is the largest and also the first group, when there is a linear trend?

Table 3 shows a significant contribution to hearing problems only in cases of Elevated BP (aged 26-35, 15-25, 46 ~ 64). Not in cases of Hypertension. Probably due to drug treatment. This must be addressed. Interestingly, self-reported hypertension is the best predictor of hearing problems. This is the cheapest variable that can be applied in early detection for the risk of hearing loss even before it has occurred.

Reviewer 2 Report

The present manuscript describes results indicating a possible relationship between hypertension and hearing problems. The study on Indonesians is based on a cross-sectional national survey 2014, IFLS5. The survey included a structured questionnaire administered through in-person interviews by well-trained interviewers and physical examination including blood pressure, body height and body weight. Hearing problems were assessed by answering a question about hearing problems. 28,297 individuals were included in the study of which 220 people had presented self-reported hearing problems. The hypertensive group comprised 12,843 participants.

Hypertension was not significantly correlated with hearing problems. However, participants with hearing problems had a significantly higher proportion of isolated hypertension than those without such problems. There were also correlations of hearing problems with being employed, had a recent general check up and having recent outpatient care. The study was also followed up by an univariate analysis of hearing problems with demographic data. Significant associations were shown between hearing problems and the oldest age group.

The study is interesting and well-written. I however have some considerations. Hearing difficulties are a major issue in this study. It is a pity that the survey did not contain audiometric measurements instead of just a question concerning hearing problems. Furthermore I cannot understand why not all data were analysed regarding males and females. It had been of interest to see if some of the correlations were more or less common in relation to gender. Despite these setbacks I am ready to recommend the manuscript for publication but not until the issue regarding gender has been analysed.

Reviewer 3 Report

Hearing problems and Hypertensions are both severe public health issues. As stated in introduction a relationship between hypertension and hearing loss is known.

Both factors increase with age. Both factors are also related to occupational noise exposure not mentioned in the paper. Occupational noise exposure is much more important for hearing deterioration.

I cannot find a specific Aim of this work. The authors state "In this study, we investigated the prevalence of hearing problems and assessed the relationship between hearing problems and blood pressure based on subjects’ age groups in an Indonesian population."

The study is based in the Indonesian Family Life Survey 5th wave. Data are cross sectional even if the IFLS two lines later is described as a longitudinal (probably meaning repeated survey. If data come from the same individuals, then you could do a cohort analysis.  I am also puzzled by the numbers given 83% of the total population. The population numbers I have says 252 000 000 inhabitants!?  The number of participants for the 5th wave are 239 806 persons.  less than 0.1%!?

It is not clear to me why the authors use self-reported and measure hypertension. The measured version categories are described in some detail line 110 -123.

Hearing problems are questionnaire based and of course suffers from reporting bias (health awareness, need for hearing including work demands, education).

The prevalence’s given are fairly low but increase with age, as expected. A surprising finding is that those with hearing aid usage do report less problems with hearing, illustrating some problems with the outcome variable.

Table 3 presents data using logistic regression with different models but I can't find a proper description of the models. The results indicate that hearing problems are increased for the older age group only and the relationship to hypertension is do not show clear dose response. The order of age groups is a little puzzling to me

I have some problems with the presentation of interactions in table 4. In my thinking they show the similar results as table 3 with addition of data of isolated systolic hypertension, indicating the same tendency. This table is not needed. The authors could perhaps focus to do a more simplified analysis restricted to the oldest age group?

The presentation with a common reference group makes it more difficult to read.

Conclusions 

I have problems with the statement with predictive value. This may be more an effect of health awareness overall. When you have measured values focus on that. I think that there might be a need for screening for both hearing disability and hypertension irrespective of any relationship. They are both of great importance for health.

I believe that the less distinctive conclusion is a product of a less clear aim of the present investigation.

 Details page numbers didn't work in my copy

Round 2

Reviewer 3 Report

line 243 I bleive you intend to write "on the other hand"

Line 358 check language here
